

# Varying soil moisture-atmosphere feedbacks explain divergent temperature extremes and precipitation projections in Central Europe

Martha M. Vogel[1], Jakob Zscheischler[1], and Sonia I. Seneviratne[1]

[1]Institute for Atmospheric and Climate Science, ETH Zurich

*Correspondence to:* Martha M. Vogel (martha.vogel@env.ethz.ch)

**Abstract.**

The frequency and intensity of climate extremes is expected to increase in many regions due to anthropogenic climate change. In Central Europe extreme temperatures are projected to change more strongly than global mean temperatures and soil moisture-temperature feedbacks significantly contribute to this regional amplification. Because of their strong societal, ecological and economic impacts, robust projections of temperature extremes are needed. Unfortunately, in current model projections, temperature extremes in Central Europe are prone to large uncertainties. In order to understand and potentially reduce uncertainties of extreme temperatures projections in Europe, we analyze global climate models from the CMIP5 ensemble for the business-as-usual high-emission scenario (RCP8.5). We find a divergent behavior in long-term projections of summer precipitation until the end of the 21st century, resulting in a trimodal distribution of precipitation (*wet*, *dry* and *very dry*). All model groups show distinct characteristics for summer latent heat flux, top soil moisture, and temperatures on the hottest day of the year (TXx), whereas for net radiation and large-scale circulation no clear trimodal behavior is detectable. This suggests that different land-atmosphere coupling strengths may be able to explain the uncertainties in temperature extremes. Constraining the full model ensemble with observed present-day correlations between summer precipitation and TXx excludes most of the *very dry* and *dry* models. In particular, the *very dry* models tend to overestimate the negative coupling between precipitation and TXx, resulting in a too strong warming. This is particularly relevant for global warming levels above 2 °C. The analysis allows for the first time to substantially reduce uncertainties in the projected changes of TXx in global climate models. Our results suggest that long-term temperature changes in TXx in Central Europe are about 20% lower than projected by the multi-model median of the full ensemble. In addition, mean summer precipitation is found to be more likely to stay close to present-day levels. These results are highly relevant for improving estimates of regional climate-change impacts including heat stress, water supply and crop failure for Central Europe.



*Copyright statement.* TEXT

# 1 Introduction

The frequency and intensity of extreme temperature events is expected to increase due to anthropogenic climate change (Christidis et al., 2011; Rahmstorf and Coumou, 2011; Seneviratne et al., 2012; Otto et al., 2012; Morak et al., 2013; Fischer and
Knutti, 2015). The occurrence and magnitude of temperature extremes strongly varies across regions and can have strong societal (e.g. Robine et al., 2008), ecological (Frank et al., 2015; Allen et al., 2010) and economic (Westerling et al., 2006; Barriopedro et al., 2011) impacts. Hence, reliable regional information for extreme temperatures and robust projections are urgently needed to develop mitigation and adaptation strategies.

Already today, a much stronger increase in extreme temperatures compared to global mean temperature can be observed
in many regions over land (Papalexiou et al., 2018), although this tendency is generally found to be smaller in observations compared to climate model simulations (Donat et al., 2017). Projections derived from simulations conducted with Earth System Models (ESMs) show a further enhancement of this regional amplification (Seneviratne et al., 2016; Wartenburger et al., 2017). However, these projections are subject to large uncertainties, particularly in mid-latitude regions such as Central Europe (e.g. Seneviratne et al., 2012; Cheruy et al., 2014). The uncertainties in climate projections arise for different reasons and it is
important to understand the underlying physical mechanisms in order to reduce uncertainties (Shepherd, 2014). In Central Europe, anticyclonic weather conditions and soil moisture drying have been identified as important drivers for the development of heat waves (Quesada et al., 2012). Over longer time scales, summer soil moisture strongly contributes to the regional amplification of extreme temperatures in climate change projections in Europe (Seneviratne et al., 2013; Vogel et al., 2017).

Soil moisture plays an essential role because it influences the partitioning of the energy available at the land surface into
the sensible and latent heat fluxes, depending on the prevailing climate regime (Koster et al., 2004; Seneviratne et al., 2010). In a transitional climate regime, evapotranspiration depends on soil moisture, which affects the surface energy fluxes and consequently temperature. This mechanism can result in a soil moisture-temperature feedback, whereby increased temperatures (e.g., due to global warming) lead to higher atmospheric moisture demand and can thus induce soil drying, which in turn can enhance the initial temperature increase (Seneviratne et al., 2010). In addition, changes in evapotranspiration may influence
precipitation via moisture input to the atmosphere, while precipitation itself also affects soil moisture (Koster et al., 2004; Seneviratne et al., 2013; Guillod et al., 2015). In present day, Central Europe is typically characterized by a wet climate regime (no soil moisture limitation) (Seneviratne et al., 2006; Teuling et al., 2009; Seneviratne et al., 2010) but can occasionally shift to a transitional regime, in particular in summer during droughts (Zscheischler et al., 2015). An example for such a regime shift was the summer 2003, during which soils were so dry that the occurring heat wave was substantially enhanced by the
lack of soil moisture (Fischer et al., 2007; Whan et al., 2015). In addition, climate projections suggest a long-term shift to the transitional climate regimes under a warmer climate, whereby soil moisture would increasingly affect summer temperature variability (Seneviratne et al., 2006, 2013; Vogel et al., 2017).




Hence, diagnosing uncertainties in soil moisture-atmosphere coupling may help to better understand uncertainties in projections of temperature extremes. Model uncertainties in the simulation of soil moisture and temperature can arise if the transition between wet and transitional regimes is not well captured (Seneviratne et al., 2006; Boe and Terray, 2008). Furthermore, varying trends in soil moisture (Lorenz et al., 2016) and systematic biases in the representation of soil moisture-temperature

feedbacks can contribute to these uncertainties (Cheruy et al., 2014; Mueller and Seneviratne, 2014; Sippel et al., 2017).

Precipitation strongly influences projected changes in soil moisture. Unfortunately changes in regional precipitation are among the most uncertain in climate change projections (Greve et al., 2017). Particularly in Central Europe, models do not agree on the sign of change (Orth et al., 2016) and correspondingly, projected changes in soil moisture are also highly uncertain (Orlowsky and Seneviratne, 2013). While there is evidence that anthropogenic climate change contributed to increasing trends

in Northern Europe and decreasing trends in the Mediterranean region, no trends are apparent in Central Europe, and reconciling observations and models remains challenging (Zhang et al., 2007; Gudmundsson and Seneviratne, 2016; Orth et al., 2016; Gudmundsson Lukas et al., 2017)

One approach to overcome these challenges and reduce uncertainties in projected changes with regard to underlying processes is the use of physically consistent observational constraints. Such constraints can be applied on multi-model ensembles

and allow the selection of the 'best' models with respect to a physically plausible metric rather than changing model code. Constraining a multi-model ensemble assumes that models which are in better agreement with a metric from present-day observed climate have a more realistic representation of relevant processes and therefore produce more reliable future projections.

However, previous studies that have applied observational constraints to projected changes in hot extremes in Central Europe come to contrasting conclusions. Christensen and Boberg (2012) performed an analysis using the global multi-model ensemble

simulations collected in CMIP5 (Coupled Model Intercomparison Project Phase 5) (Taylor et al., 2012). They show that models tend to have a warm season bias in regions where land-atmosphere feedbacks are important (Christensen and Boberg, 2012) such as Central Europe. Borodina et al. (2017) suggested that uncertainties in projections of the hottest day of the year (TXx) are linked to present-day climatology and concluded that the frequency of hot extremes are likely to increase at a higher rate than the multi-model estimate for large parts of the northern hemisphere based on a TXx scaling constraint (assuming constant

TXx increase with summer mean temperature increase). However, they did not find a robust signal for Central Europe. Sippel et al. (2017) applied a land-surface coupling metric (Zscheischler et al., 2015) to CMIP5. Their results suggest that temperature extremes in Central Europe are likely to be lower than predicted by the multi-model mean, but the applied constraint has little effect on the change in temperature extremes in a warmer climate (Sippel et al., 2017). Hence, the question of the extent to which temperature extremes in Central Europe are projected to increase under enhanced greenhouse forcing and whether these

projections can be substantially constrained with observations still remains to be answered.

In this study we investigate projected changes over summer in Central Europe in the CMIP5 ensemble in order to better understand the large uncertainties in projected changes of TXx. We investigate underlying mechanisms in ESMs which are relevant for changes in temperature extremes. In the first part we identify dominant processes in the models which can explain uncertainties in TXx projections. We focus on the role of land-atmosphere interactions by investigating the relationship between

land surface and atmospheric variables during summer, in particular precipitation, latent heat flux, soil moisture and TXx. The





analysis further motivates the usage of a process-based constraint that quantifies the strength of land-atmosphere coupling. To this end, we use the correlation between summer precipitation and TXx. Applying this constraint allows us to substantially reduce uncertainties associated with projections in summer precipitation and temperature extremes in Central Europe.

## 2  Data and Methods

### 2.1  CMIP5

We investigated 23 state-of-the-art climate models with up to 10 ensemble members (Table 1) from the CMIP5 archive (Taylor et al., 2012) for the historical period and the high-emissions scenario RCP8.5 (Meinshausen et al., 2011). The RCP8.5 scenario exhibits the strongest warming signal at the end of the 21$^{st}$ century and thus has a high signal-to-noise ratio to detect robust changes. We used all models and ensemble members that were available for our considered variables (see below), resulting in a total of 44 model realizations. The choice to use all available realizations was made because we found that the intra-model variability is similar to the inter-model variability for the investigated variables.

We analyzed changes over land in precipitation, latent heat flux, top soil moisture, radiation, and TXx. We calculated TXx from daily maximum temperature data at each grid box and for each model. The resulting TXx values occur on different days at different locations in different models. From the resulting TXx fields we computed area-weighted averages across the SREX region Central Europe (CEU, see inset in Figure 1). For all other variables we calculated summer means and averaged across CEU.

For each variable we studied changes between 1950 and 2100. As our focus is on long-term trends, we calculated 20-year means to remove interannual variability. The years indicated for timeseries in the plots are the center of the 20 years (year 11). Changes are calculated as differences to the base period 1950-1969. This allowed to exclude model bias and directly compare long-term trends in model runs. For the distributions at the end of the 21$^{st}$ century (see figures) we compared means of 2081-2100 with this base period 1950-1969.

We present changes over time also relative to changes in global mean temperature, following the near-linear relationship between cumulative $CO_2$ emissions and global mean temperatures (IPCC, 2013). We estimated global mean temperature (Tglob) as the average of all 44 models. We then calculated 20-year means and computed changes from 1951-2100 with respect to the base period 1950-1969. To account for changes with respect to pre-industrial levels we added the multi-model mean increase from the 44 models of the CMIP5 ensemble from 1871-1890 to 1950-1969 to the changes (0.23 °C).

### 2.2  GLACE-CMIP5

We further made use of the output from five ESMs that contributed to the GLACE-CMIP5 Experiment (Seneviratne et al., 2013) to understand the role of soil moisture-temperature feedbacks in climate-change projections. We analyzed two experiments, the CMIP5-like reference simulation (hereafter referred to as GLACE CTL) and the simulations with prescribed 20$^{th}$ century soil moisture conditions to suppress the impact of soil moisture-climate feedbacks in the projections (SM20c in Vogel et al. (2017),





hereafter referred to as GLACE SM20c). The GLACE SM20c experiment removes the projected long-term drying of soil moisture as well as the short-term soil moisture variability. When comparing GLACE SM20C and GLACE CTL, differences in climate are thus due to the removed soil moisture trend and the removed short-term soil moisture-climate interactions. All simulations cover the time period 1951–2100 using historical forcing until 2005 and forcing from the RCP8.5 scenario from

2006-2100 (Meinshausen et al., 2011). Since GLACE-CMIP5 simulations are available only from 1951 (not 1950) we adjusted the base period to 1951-1970 when considering the GLACE-CMIP5 experiments.

## 2.3   Observations

We used gridded data for TXx and summer precipitation from E-OBS (version 15, Haylock et al., 2008), CRU version 4.1 (Harris et al., 2014), GPCC version 7 (Schneider et al., 2014, only precipitation), and HadEX (Hadley Centre Global Climate

Extremes Index 2, only TXx) from Donat et al. (2013), which are derived form station observations. Furthermore we used Princeton forcing data (Sheffield et al., 2006) and GWSP3 (the third Global Soil Wetness project updated from Dirmeyer et al. (2006)), a global meteorological forcing dataset used as forcing for the CMIP6 experiments. All data are available for the reference time period 1961-1990 (which was established in the IPCC AR5) and were area-weighted averaged over CEU to compare with model output.

## 3   Results

### 3.1   Projected increase in TXx and divergent changes in summer precipitation

TXx increases for all 44 model realizations in CEU until the end of the 21$^{st}$ century (Figure 1a). The multi-model median is increasing by around 9.5°C until the end of 21$^{st}$ century (Table 1) with values ranging from 3 to 13°C, which is in agreement with results from various other studies (e.g. Seneviratne et al., 2012; Vogel et al., 2017). At the end of the 20$^{th}$ century, the

multi-model median of precipitation shows no clear trend and changes of the individual models differ between -0.3 and 0.2 mm/day (Figure 1b). At the beginning of the 21$^{th}$ century, model differences increase and model runs start diverging with respect to precipitation changes. Because the majority of the models show decreasing precipitation at the end of the 21$^{st}$ century, the multi-model median is decreasing to -0.44 mm/day. Kernel density estimates suggest a trimodal distribution of summer precipitation changes at the end of the 21$^{st}$ century (Figure 1b). We use these density estimates to classify the model

runs by selecting the two local minima of the trimodal distribution as the boundaries for the three groups. Table 1 shows which model run was assigned to which group. 10 models are within the first mode, associated mostly with positive changes of summer precipitation and are referred to as *wet* hereafter (blue in Figure 1). 18 models are in the middle range of the precipitation distribution, associated with a slight drying in CEU at the end of the 21$^{st}$ century, referred to as as *dry* (orange in Figure 1). 16 models are in the lower tail of the distribution associated with a strong decrease of precipitation up to more than

1 mm/day, which are referred to as *very dry* models (red in Figure 1). The multi-model median for projected changes at the end of the 21$^{st}$ century between these three groups differs strongly from 0.13 mm/day for the *wet* ensemble to -0.36 mm/day





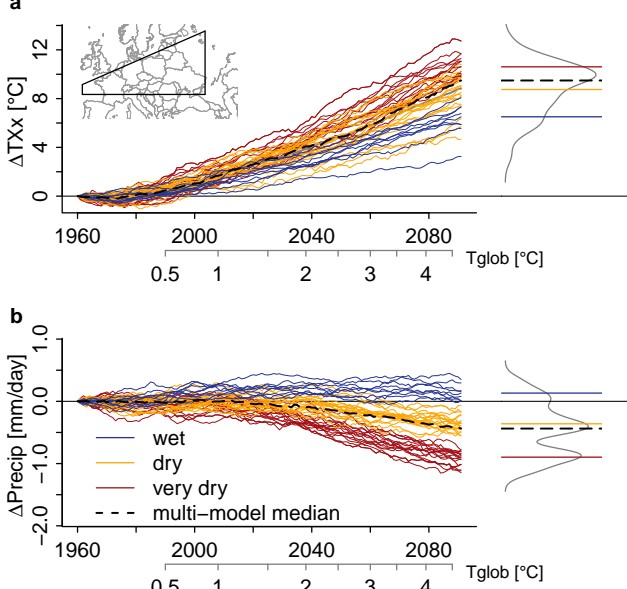

**Figure 1.** Change in a) TXx and b) summer precipitation (precip) in Central Europe for 44 model realizations for wet (blue), dry (orange), and very dry (dark red) models and the multi-model median (dashed). Changes are calculated as 20-year running means with respect to the base period 1950-1969. Density distributions are shown for changes in 2081-2100 with respect to the base period (right). The horizontal lines in the density distributions indicate the multi-model median of the wet (blue), dry (orange), very dry (darkred) ensemble and the multi-model median (dashed) for changes in the 2081-2100 period.

for the *dry* ensemble to a decrease of -0.90 mm/day for the *vey dry* ensemble. The median of the whole ensemble is within the *dry* ensemble (Table 1).

Applying the classification to TXx, we find that hottest models tend to be *very dry* whereas *dry* and *wet* models show less strong increases in TXx, even though the distribution of TXx does not show a clear trimodal behavior (with the *dry* and *wet*

5    models partly overlapping). The medians of TXx of the *wet*, *dry* and *very dry* ensembles are ranging from 10.6 °C to 8.7 °C and 6.5 °C, respectively (Table 1), implying a difference between the median of the *wet* and *very dry* models at the end of the 21$^{st}$ century of more than 4 °C (Figure 1a).

The clustering of the three model groups according to precipitation trends being partly reflected in the ensemble of the TXx trends (at least for the *very dry* vs. *dry* and *wet* models) suggests a critical role of land-atmosphere interactions. Precipitation

10    can be seen as a proxy for dryness and a decrease in precipitation decreases soil moisture, influencing the partitioning of surface heat fluxes and consequently air temperature. The change in precipitation may be driven by large-scale circulation and local processes such as soil moisture-precipitation feedbacks and convection. To examine these relations in more detail, we investigate other summer variables in the following section.



**Table 1.** Classification of CMIP5 models in three subgroups. Changes of the multi-model median of TXx, precipiatation (precip), latent heat flux (LH), top soil moisture (Top SM), incoming shortwave radiation (SWin) and net radiation (Rnet) are shown between 2081-2100 and 1950-1969. The number in brackets corresponds to the number of ensemble members. If not indicated then only one ensemble member is used.

| Subgroup | Model | $\Delta$ TXx | $\Delta$ Precip | $\Delta$ LH | $\Delta$ Top SM | $\Delta$ SWin | $\Delta$ Rnet |
|---|---|---|---|---|---|---|---|
| | | °C | mm/day | W/m$^2$ | kg/m$^2$ | W/m$^2$ | W/m$^2$ |
| *Wet* | CNRM-CM5, FGOALS-g2, IPSL-CM5A-LR[1] MIROC5 (2), MIROC-ESM-CHEM MIROC-ESM, MRI-CGCM3 | 4.0 | 0.1 | -11.9 | -1.0 | 21.5 | 22.3 |
| *Dry* | ACCESS1-3m, bcc-csm1-1 CanESM2 (5), CCSM4 (2) CESM1-BGC, GFDL-CM3, GFDL-ESM2G GFDL-ESM2M, inmcm4 IPSL-CM5A-LR (3) IPSL-CM5A-MR, NorESM1-M | 8.7 | -0.4 | -1.8 | -2.0 | 22.1 | 19.3 |
| *Very dry* | ACCESS1-0 CSIRO-Mk3-6-0 (10) HadGEM2-CC, HadGEM2-ES (4) | 10.6 | -0.9 | -16.8 | -4.0 | 43.0 | 18.7 |

[1] r4i1p1

## 3.2   The role of changes in summer land and atmosphere variables

We analyze changes in latent heat flux, incoming shortwave radiation, net radiation, convective and stratiform precipitation, and top soil moisture rather than total soil moisture as we only expect a strong exchange with the atmosphere in the upper layers of the soil (Cheng et al., 2016). All models show a decrease in top soil moisture, with a clear clustering following the

5 three identified model subgroups (Figure 2). *Wet* models only show a slight decrease of around 1 kg/m$^2$, whereas *very dry* models show a strong soil moisture decrease of 4 kg/m$^2$, with *ACCESS1-0* exhibiting the strongest decrease (Table 1). Summer latent heat flux clusters similarly to precipitation, following a trimodal distribution at the end of the 21$^{st}$ century. We note that the divergence of the three subgroups starts at a similar time as for precipitation. *Wet* models show a continuously increase of latent heat flux, *dry* models show an overall slight decrease of latent heat flux and the *very dry* models show a strong decrease

10 of latent heat flux until 2100 (Figure 2b). All models show and increase in incoming shortwave radiation in summer. We find strongest increases for the *very dry* models. However, the *dry* and *wet* models do not show a distinguishable behaviour, the medians for these two groups are similar and the distributions overlap strongly (Figure 2c). No detection of three groups is possible for net radiation (Figure 2d). Two of the *wet* models (*MRI-CGCM3, MIROC-ESM*) show the strongest increase of net




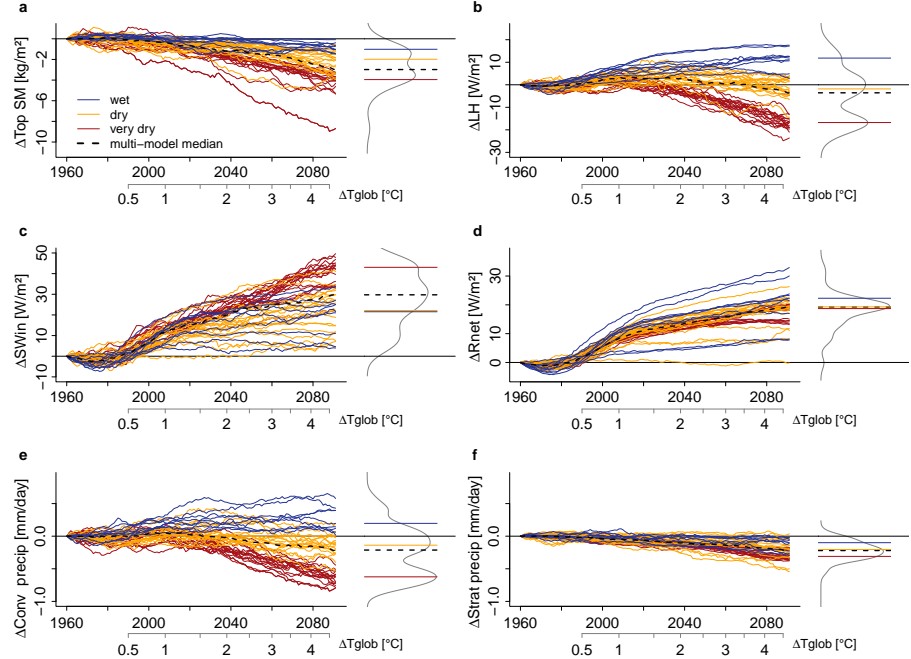

**Figure 2.** Change in summer a) top soil moisture (Top SM), b) latent heat flux (LH), c) incoming shortwave radiation (SWin), d) net radiation (Rnet), e) convective precipitation (conv precip) and f) stratiform precipitation (strat precip) in CEU for *wet* (blue), *dry* (orange) and *very dry* models (red), the multi-model median (dashed). Changes are calculated as 20-year running means with respect to the base period 1950-1969. Density distributions are shown for changes between 2081-2100 and the base period.

radiation even though their TXx increase is rather small. Hence medians of the three model subgroups are very similar at the end of the 21$^{st}$ century. (Note that *incm4* does not show an increase in summer incoming shortwave radiation over CEU.)

To understand the causes for the precipitation decrease, we analyse convective and stratiform precipitation separately. We exclude *CCSM4* because of artifacts in the partitioning of precipitation. The evolution of convective precipitation is very similar

5  to that of total precipitation and we find again *wet*, *dry* and *very dry* models (Figure 2e). In contrast, the stratiform summer precipitation is overall slightly decreasing until 2100 and the three model subgroups are not distinguishable (Figure 2f). We also considered changes in geopotential height (500hPa) and could not find systematic behaviours in the models (not shown).

Overall these timeseries allow us to identify two phases: i) Until the beginning of the 21$^{st}$ century: An increase in net radiation associated with increases in latent heat flux and TXx rather independently of any changes in soil moisture and precipitation.

10  ii) Afterwards: Evolution of the the divergent behavior for precipitation resulting in a trimodal distribution. The changes of the variables for the three model subgroups are summarized in Table 1. The three groups can be characterized as follows: a) *Wet* models tend to show further increase in net radiation with only little decrease of soil moisture, associated with an increase in precipitation, increase in latent heat flux and less strong increase of TXx around 6 °C for the median of the *wet* models 1. b) *Dry* models show less strong increase in net radiation, decrease in soil moisture associated with a reduction in precipitation and





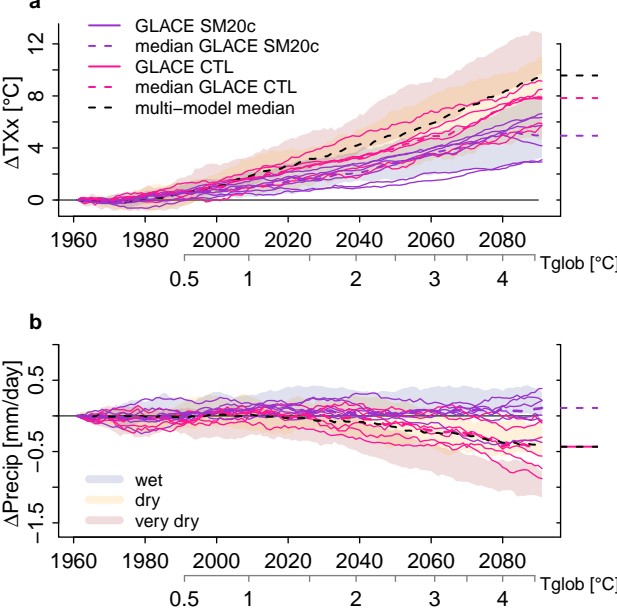

**Figure 3.** Change in a) TXx and b) summer precipitation (precip) in CEU with respect to the base period 1951-1970. Changes are computed based on 20-year moving averages with respect to the base period. The shaded area shows the minimum and maximum from the *wet* (blue), *dry* (orange) and *very dry* (red) models. Changes for GLACE CTL (violet) and GLACE SM20c (pink) are shown. Dashed lines represent the multi-model median of the full ensemble (black) and GLACE CTL (violet). Density distributions are shown for changes during 2081-2100 (right) for the GLACE CTL (violet) and GLACE SM20c (pink).

latent heat flux and strong increase in TXx of more than 8 °C for the median of the *dry* ensemble. c) *Very dry* models display a similar increase of net radiation as the *dry* models but a stronger decrease of soil moisture associated with stronger decrease of precipitation, latent heat flux and strongest increase of TXx (more than 10 °C for the multi-model median).

The *very dry* models are characterized by a strong link between precipitation, latent heat flux and TXx, and net radiation
5    might not be the only driver for the strong increase in TXx. In the *wet* models, net radiation increase might increase latent heat flux, which might increase precipitation and therefore decrease top soil moisture only slightly and consequently lead to a less strong increase for TXx.

### 3.3 Soil moisture as driver for divergent summer precipitation in models?

The previously presented results suggest a strong contribution of land-atmosphere interactions for projected changes in TXx.
10    However, from the time series we can only hypothesize on the underlying mechanisms. Therefore, for a more in-depth understanding of the role of soil moisture as possible driver for precipitation divergence we analyze GLACE-CMIP5 model simulations (Seneviratne et al., 2013). In GLACE SM20c, the soil moisture-climate feedbacks are switched off and there is typically more water available in the model simulations as soils are not drying in comparison to GLACE CTL. Both for TXx





and precipitation, the GLACE CTL runs are within the range of the full CMIP5 ensemble. In particular for precipitation the median of GLACE CTL and the multi-model median of CMIP5 are equal, showing a decrease to -0.4mm/day at the end of the 21$^{st}$ century (Figure 3b). The warming in GLACE CTL is around 1.8 °C weaker at the end of the 21$^{st}$ century than in the full CMIP5 ensemble (7.8. °C vs 9.6 °C). The GLACE CTL and GLACE SM20C simulations show strong differences in the

projected increase of TXx and precipitation. The GLACE SM20c simulations are associated with less strong warming and only show an increase of TXx of 4.9 °C at the end of the 21$^{st}$ century. All but one GLACE SM20C simulations show an increase in summer precipitation (Figure 3a) resulting in 0.1 mm/day at the end of the 21$^{st}$ in contrast to -0.4 mm/day for GLACE CTL and the full ensemble. Hence for GLACE SM20c changes in summer precipitation are shifted towards *wet* conditions. This suggests that soil moisture-precipitation feedbacks strongly contribute to the drying precipitation signal found in the *dry* and

*very dry* models.

## 3.4 Constraining: Which pathways are more realistic in projections?

After assessing relevant relationships and processes in the three identified model groups, we investigate whether a *wet*, *dry* or *very dry* pathway is more likely in the future and therefore compare our results to observations. We focus on precipitation and TXx, as we identified a link between these variables in the models, and also because well constrained gridded observations are

available for CEU. We show changes in precipitation and TXx for five different datasets described in Section 2.3. For the length of the observational time period, trends in TXx and precipitation are within the range of the model estimates (see Figure 4). We find a very similar evolution of TXx between HadEX and EOBS as well as between CRU, GSWP3 and Princeton, which is probably related to sharing the same underlying data. Overall, the datasets show a decrease of TXx from 1960 onward and an increase only after 1980. This evolution might be the result of aerosol affects and global dimming and brightening (Wild et al.,

2005; Sanchez-Lorenzo et al., 2015).

Summer precipitation shows a stronger variability. CRU shows most of the time a slight decrease whereas the other datasets are slightly increasing in the 1970s and decreasing after 1980. This could be again related to effects of global dimming and brightening. Until 1990 GPCC, GSWP3 and Princeton show very similar changes in precipitation indicating that the forcing datasets were using the same precipitation. Most of the CMIP5 models do not show the dimming and brightening evolution

of precipitation and TXx. However, after 1990 observed TXx and precipitation are close to to the multi-model median. We conclude that considering univariate timeseries will not help to reduce uncertainty.

Suspecting a coupling between precipitation and TXx, we compute the spatially averaged correlation of precipitation and TXx (cor(TXx, precip)) for present (1961-1990) and future (2071-2100). Such a correlation-based metric is commonly used to diagnose land-atmosphere coupling (Seneviratne et al., 2006; Lorenz et al., 2012; Miralles et al., 2012). The correlation

cor(TXx, precip) is always negative and varies largely across models (between -0.64 and -0.19 for present-day) but seems to be a model feature that is fairly consistent through time, resulting in a correlation of R=0.74 (p<0.001) between present-day and end-of-century correlation (cor(TXx, precip)) across models (Figure 5). We determine the observed range by the minimum and maximum values from the total of five correlations based on observational products described in Section 2.3. The observations cover a rather small range (between -0.45 and -0.28) and corresponds to the medium to upper range of the models (Figure 5).





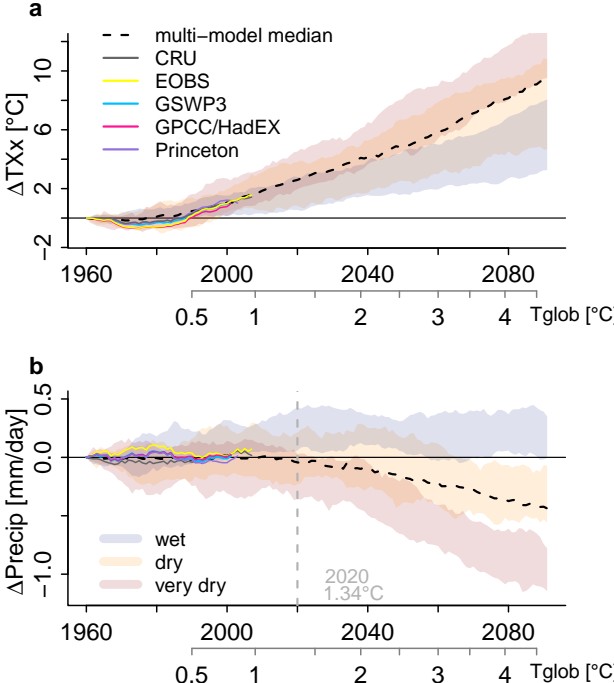

**Figure 4.** Changes in a) TXx and b) summer precipitation (precip) in Central Europe. The multi-model mean median (dashed black) of the whole ensemble is shown. The shaded area shows the minimum and maximum from the *wet* (blue), *dry* (orange) and *very dry* (red) models. The gray lines show changes of CRU, EOBS, GSWP3, GPCC/HadEX, Princeton (20-year means) from 1950-1969 until the end of the observed time periods which ranges from 2010 to 2016. The gray vertical line indicate from where the distributions of precipitation for the *very dry* and *wet* models do not overlap.

Most of the *very dry* and *dry* model runs can be excluded from the multi-model ensemble. The *constrained* model ensemble includes 13 models, mainly from the *wet* and *dry* ensemble (Figure 5). The projected distributions for TXx and precipitation show a substantial reduction in model spread (Figure 6a). The spatial pattern show a strong reduction in TXx and increase in precipitation compared to the full ensemble (Figure 7). Interestingly, the multi-model median of the *constrained* ensemble

5    hardly shows a change in precipitation (-0.17 mm/day compared to -0.43 mm/day for the full ensemble, Figure 6b). For large parts of CEU no change in the precipitation trend is detected (Figure 7). Particularly the dry and the hot tails of the projected distributions are removed, resulting in a reduction in TXx by around 2 °C (from 9.5 °C to 7.5 °C) which corresponds to a reduction by 20 %. The *constrained* ensemble indicates less strong drying since models with very strong decrease in top soil moisture and latent heat flux are removed (Figure 6c & d). Soil moisture is only projected to increase by 1.45 kg/m$^2$

10   which reduces the projected drying for the full ensemble by 50%. The *constrained* multi-model median of latent heat flux changes sign, projecting an increase of 2.8 W/m$^2$ in contrast to a decrease of -3.6 W/m$^2$ for the full ensemble. Uncertainties of the projections are are slightly reduced due to the less likely dry tails of the distribution (Figure 6). These results suggest that





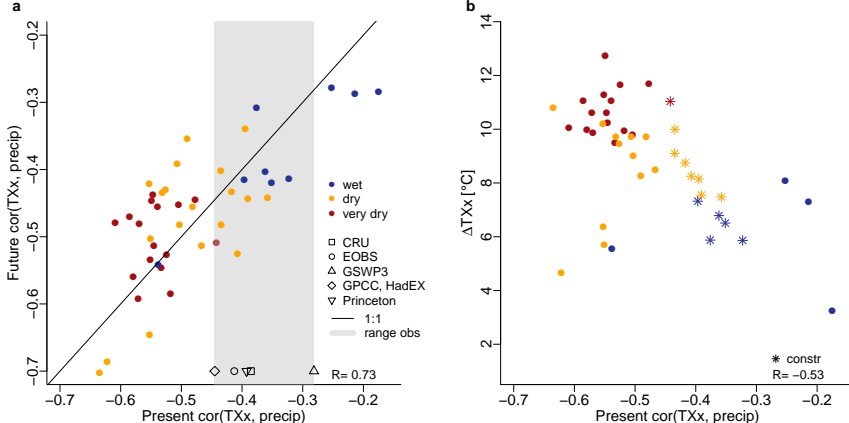

**Figure 5.** a) Future versus present-day cor(TXx, precip) for *wet* (blue), *dry* (orange), *vey dry* models (red) and for observations. The 1-to-1 line is shown in black. The gray background depicts the minimum and maximum from the distribution of cor(TXx, precip) of the observational datasets. b) Projected incrase in TXx versus present-day cor(TXx, precip). The stars indicate the models within the constrained ensemble, the colors refer to the three model subrgoups as in a).

observationally constrained long-term changes in summer TXx in CEU are within the lower range of the multi-model ensemble and associated with only little decrease in summer precipitation.

Furthermore, we find that the distributions of the full and *constrained* ensemble are still very similar for global warming levels of for 1.5 and 2 °C, but show strong differences for 3 and 4 °C (Figure 8). This indicates that the model uncertainties only play a major role at high warming levels. Applying a scaling of TXx and precipitation with Tglob as in Seneviratne et al. (2016) we find a reduction for TXx of 1 °C and for precipitation of 0.3 mm/day for the *constrained* compared to the full ensemble for global warming of 4.5 °C which corresponds to a reduction of TXx increase from 1.8 °C/°C Tglob to 1.6 °C/°C Tglob (Figure A2). For top soil moisture and latent heat flux the inter-quartile range of the *constrained* is shifted towards wetter conditions for 3 °C and 4 °C Tglob increase (Figure A1). Overall *very dry* and hot projections are excluded for global mean temperature increase above 2 °C (Figure A1).

## 4 Discussion

### 4.1 Feedbacks

In our study we identify *wet*, *dry* and *very dry* models for CEU with distinct characteristics for latent heat flux, soil moisture and TXx (Table 1), indicating the importance of the interactions between land and atmosphere in ESMs. The different characteristics hint towards systematic differences in models with respect to these land-atmosphere feedbacks.





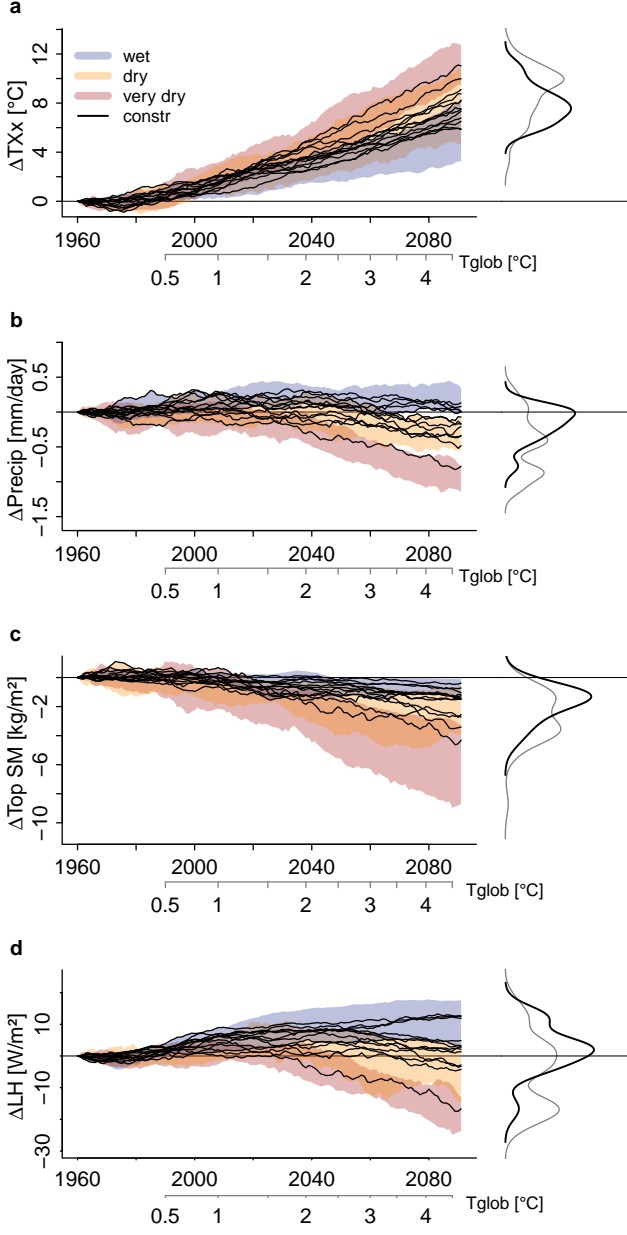

**Figure 6.** Changes in a) TXx, b) summer precipitation (precip), c) top soil moisture (Top SM), d) latent heat flux (LH) in Central Europe. Changes depicted as 20-year moving averages relative to the base period 1950-1969. The shaded areas cover the minimum and maximum from the *wet* (blue), *dry* (orange) and *very dry* (red) models, respectively. The green lines show the changes of individual model runs of the constrained ensemble. Density distributions are shown for changes during 2081-2100 for the whole (grey) and the constrained (green) ensemble (right).



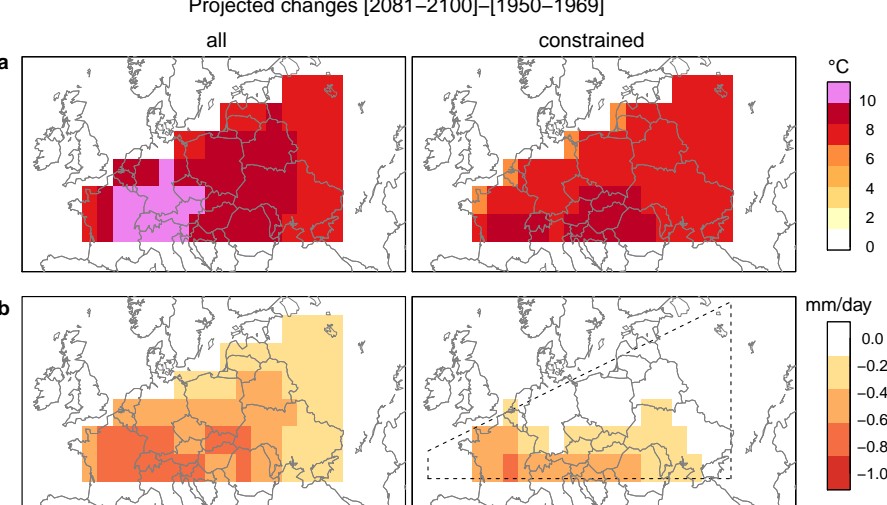

**Figure 7.** Future versus present-day a) TXx (top) and b) summer precipitation (bottom) of the multi-model mean from the full (left) and the constrained (right) ensemble.

Our analysis suggests that there are mainly three positive feedbacks which are relevant for uncertainties in TXx projections in the multi-model ensemble: 1) the soil moisture-temperature feedback, 2) the soil moisture-precipitation feedback, and 3) the soil moisture-radiation feedback (Figure 9).

To understand feedback (1) we investigate model experiments from the GLACE-CMIP5 ensemble with and without pre-
scribed soil moisture (GLACE CTL and GLACE SM20c). The difference between GLACE CTL and GLACE SM20C only results from suppressing soil moisture-climate feedbacks (Figure 3). A drying of soils leads to a decrease of latent heat flux associated with an increase of temperature, which in turn further enhances latent heat flux and thus further decreases soil moisture (Figure 9, red). In particular, the time series of the *very dry* models suggest that a strong soil moisture-temperature feedback enhances TXx (Figures 1 & 2). This is in agreement with several studies that have shown that in CEU the particular
strong temperature increase of hot extremes is mostly related to soil moisture-temperature feedbacks (Seneviratne et al., 2006; Fischer et al., 2007; Diffenbaugh and Ashfaq, 2010; Whan et al., 2015; Lorenz et al., 2016; Vogel et al., 2017). Donat et al. (2017) showed that an increase in TXx is associated with an increase in sensible heat and decrease in latent heat flux at the specific day when the hot extreme occurs, which is associated with soil moisture drying. In particular, the projected drying trends in soil moisture lead to an increases in intensity, frequency, and duration of temperature extremes by the end of the 21st
century (Lorenz et al., 2016).

In addition to this first feedback (1), we identify feebdack (2), the soil moisture-precipitation feedback, as relevant driver for TXx uncertainty in the model ensemble. Under moisture limitation, a soil moisture increase leads to a latent heat flux increase, thus cloud cover increase and result in an increase in precipitation which further increases soil moisture (Figure 9, blue). This feedback can dampen TXx via the 1) the soil moisture-temperature feedback, if advection is negligible. We find





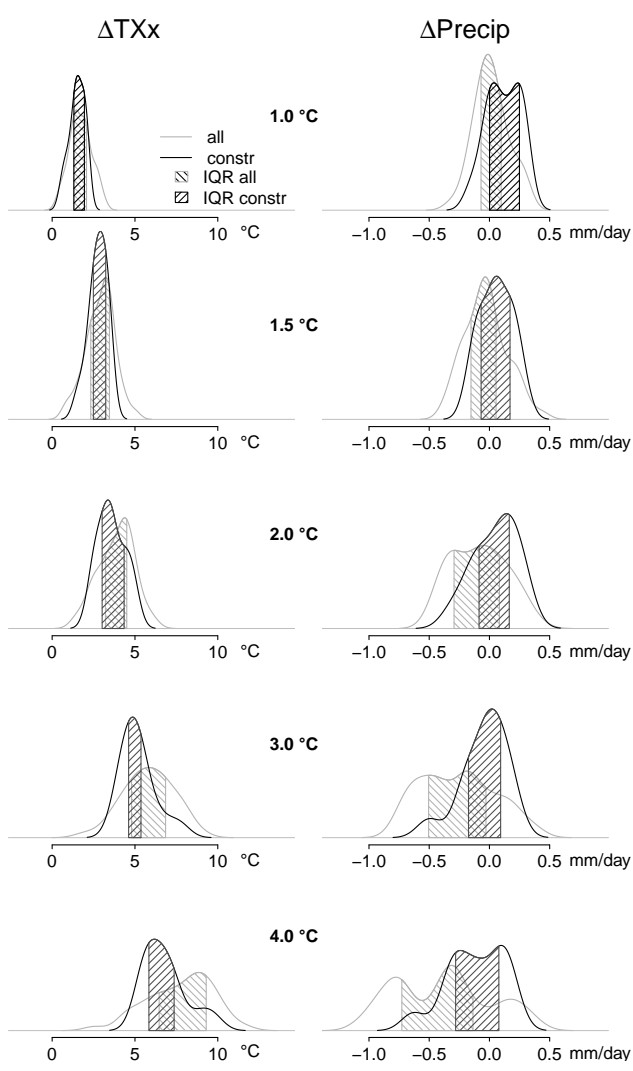

**Figure 8.** Distribution of TXx (left) and precipitation (right) changes according to global warming levels of 1 °C, 1.5 °C, 2 °C, 3 °C and 4 °C for the full (grey) and the constrained ensemble (black). The shaded area represents the interquartile range (IQR) of the two distributions.



in particular that the *wet* models show an increase in latent heat flux and precipitation, and less strong increases in TXx. The GLACE-CMP5 experiments support the hypothesis that this feedback plays and important role in determining the magnitude of the trends in TXx and reveal the relevance of moisture recycling in the models (Figure 3). If soil moisture in summer is drying over CEU due to increased atmospheric demand associated with warmer temperatures in a future climate, this feedback

mechanisms can amplify the increase in TXx as discussed above for the *very dry* and *dry* models. A measure for the effect of (2) on temperature is the correlation of precipitation and TXx. Multiple studies highlighted that the dominant pathway for negative correlations between seasonal temperature and precipitation is through the direct control of soil moisture on surface heat flux partitioning (Trenberth and Shea, 2005; Berg et al., 2015; Zscheischler and Seneviratne, 2017). Our result show that this correlation strongly influences future projections of hot extremes. The most negative present day correlations show

strongest warming for TXx (Figure 5a). However, the influence of initial changes in precipitation on soil moisture cannot be studied with this setup and ESM model experiments with prescribed precipitation are not available. We expect in addition to strong effects from soil moisture changes a causal relationship from precipitation to soil moisture.

Furthermore, we identify that feedback (3), the soil moisture-radiation feedback (via changes in cloud cover), additionally effects TXx. A decrease in soil moisture decreases latent heat flux, which can decrease cloud cover and enhance incoming

shortwave radiation. This can directly decrease latent heat flux and also decrease soil moisture via an increase in temperature and latent heat flux (Figure 9, yellow). We find that particularly *very dry* models show a strong increase in shortwave radiation at the surface in summer. This is likely related to a decrease in cloud cover. Clouds can reflect shortwave radiation at the top of the atmosphere and thus less shortwave radiation reaches the ground. Net radiation does not increase more strongly in *very dry* models, indicating that incoming shortwave radiation is not caused by an overall increase in net radiation. These considerations

are in agreement with studies showing a significant decline in cloudiness over Europe, associated with an increase in solar radiation at the surface (Wild et al., 2015; Bartók et al., 2017). Bartók et al. (2017) also stated that this decline might be related to a drying in summer over Europe, limiting the amount of water available for cloud formation. In addition, cloud formation strongly depends on aerosols. Interestingly, most of the GCMs from CMIP5 underestimate the "brightening" over Europe, likely because of the inappropriate trends in aerosol atmospheric content (Cherian et al., 2014). This would explain the

difference in temperature trends between ESMs and observations. On the other hand, a positive trend in incoming shortwave radiation over Europe is likely the result of declining aerosol burdens (Wild et al., 2015).

The constrained ensemble indicates that a very strong increase of incoming shortwave radiation is less likely since we exclude most of the *very dry* and *dry* models. This would support the findings of Wild et al. (2013), who demonstrated that CMIP5 models tend to overestimate incoming solar shortwave radiation, which is consistent with an underestimation of low

and mid-level clouds (Zhang et al., 2005).

Overall, our results suggest that the three feedbacks illustrated in Figure 9 considerably contribute to uncertainties in TXx projections as the representation of processes that govern these three feedback mechanisms may largely differ across models. We show that the divergent behavior in precipitation projections, associated with divergent behavior in latent heat flux and different drying pathways of soil moisture can explain trends and large uncertainties in TXx. This reveals that thermodynamical

aspects associated with climate change play a major role for determining changes in temperature extremes in Europe.



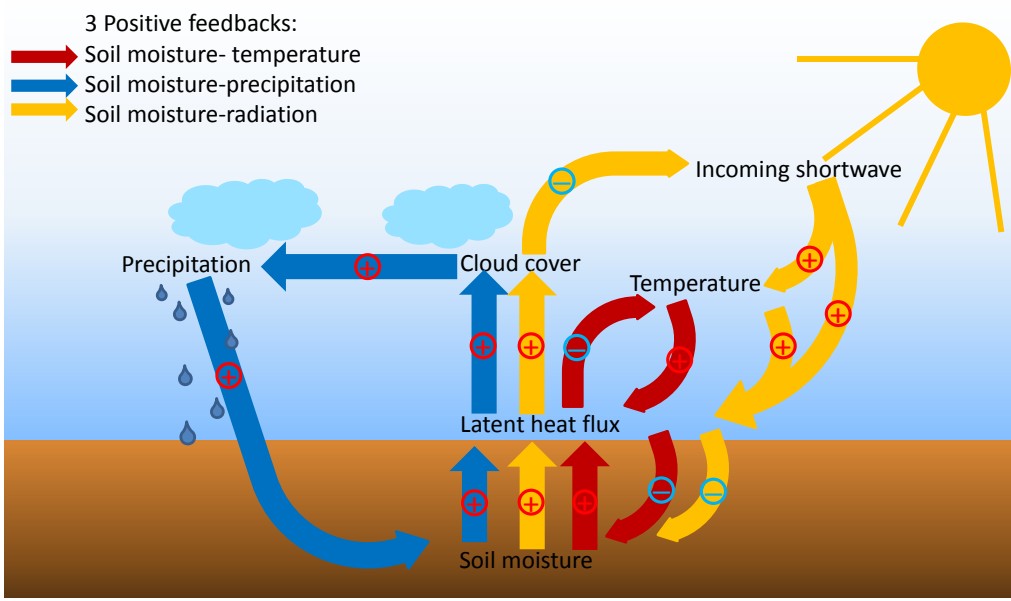

**Figure 9.** Soil moisture-atmosphere feedbacks. The plus and minus indicate positive and negative feedback loops respectively. The colors show the soil moisture-temperature (red), the soil moisture-precipitation (blue) and the soil moisture-radiation (yellow) feeedback loop.

## 4.2 Role of dynamics

Previous studies have shown that in addition to soil moisture drying, the persistence of blockings is essential for the development of heatwaves (Fischer and Schär, 2010; Pfahl and Wernli, 2012; Quesada et al., 2012; Miralles et al., 2014). Hence, changes in large-scale circulation features may also influence projected changes of temperature extremes. A recent study

suggested that the observed increase in extreme summer heat over Europe is attributable to both an increasing frequency of blockings and changes thermodynamics (Horton et al., 2015). However, CMIP5 models also exhibit large biases in blocking frequency and underestimate blocking particularly over Europe in winter but also in summer (Scaife et al., 2010; Anstey et al., 2013). The uncertainty of present climate may then be transferred to future projections, where models potentially disagree on changes in circulation-related variables in many regions (Shepherd, 2014). A case study for Australia has shown that uncertain-

ties in the climatological frequency of blockings can cause uncertainties in the transition and persistence of them (Gibson et al., 2016). Therefore, reducing uncertainties in large-scale circulation patterns in climate models might be a promising avenue to reduce uncertainties in temperature extremes.

    When analyzing the partitioning of precipitation in convective and stratiform precipitation, we find large changes in convective precipitation as well as in the clustering of the three identified subgroups. This indicates that the divergent model behavior

is linked to local convection rather than changes in large-scale circulation. To further investigate the role of atmospheric dynamics we analyzed changes in geopotential height (500hPa) in summer and could not identify any systematic differences between the different model groups. These results complement findings from Teng et al. (2016) who showed that projected



changes of heat waves in the US are primarily caused by local land-atmosphere feedbacks and not by changes in atmospheric circulation (i.e., planetary wave variability). Overall, the present analyses and inferences from reviewed literature suggest a dominant role of local land-atmosphere feedbacks for projected changes of TXx in CEU rather than changes of dynamics.

### 4.3 The choice of the constraint

When applying cor(TXx, precip) as bivariate process-based metric, we derive a *constrained* ensemble which suggests a less strong projected drying and temperature increase than for the multi-model median of the whole ensemble. The uncertainties in TXx are strongly reduced (about 20% less strong increase compared to full model ensemble) an we can particularly exclude very hot and *very dry* models.

However, our results depend on i) the choice of the constraint itself ii) the quality of the underlying observations and iii) the
criterion to determine the range for the model selection.

i) We derive cor(TXx, precip) as constraint as our analysis shows a important relationship between summer precipitation and TXx. We choose 1961-1990 as present-day period which is commonly used and does not include too strong warming trends observed after 2000. For the future period we select the last 30 years from the projections, 2071-2100. The overall negative correlations between summer temperatures and precipitation can be explained with soil moisture-atmosphere coupling and
is a well-known feature of terrestrial climate (Madden and Williams, 1978; Trenberth and Shea, 2005; Berg et al., 2015; Zscheischler and Seneviratne, 2017). Future and present correlations show a relation (R=0.73, p-Value<0.001) which makes the coupling to be a model intrinsic characteristic (Figure 5a). Furthermore, the strength of this correlation is associated with the magnitude of TXx increase (Figure 5b). In contrast to using only a single variable for constraining, this metric captures the precipitation-temperature coupling as process-based constraint. A bivariate correlation-based metric has been used frequently
in the past to test and investigate land-atmosphere coupling (Seneviratne et al., 2006; Hirschi et al., 2011; Lorenz et al., 2012; Miralles et al., 2012).

To ensure ii), we use five state-of-the-art gridded observational datasets for precipitation and TXx, which provide sufficiently long and high-quality information for CEU. EOBS, CRU, HadEX (only TXx) and GPCC (only precipitation) are based on station observations, while the Princeton forcing and GSWP3 are high-quality forcing datasets for land surface models. These
datasets are well established and continuously updated. GWSP3 is the newest forcing dataset, which will be used to drive the next generation of offline land surface simulations as part of CMIP6, in the context of the Land Surface, Snow, and Soil Moisture Model Intercomparison Project (LS3MIP, van den Hurk et al., 2016).

iii) The observed range is dependent on the available observations and choice of the time period. We tested the sensitivity to changing time periods and dataset, which did not qualitatively affect the conclusions of our study. To define the range, we
compute the minimum and maximum correlation of the five datasets. When using a smaller/larger threshold we would select less/more models which would influence the reduced uncertainties but results remain qualitative similar and we can exclude in particular *very dry* and hot models.



## 4.4 Future projections in Central Europe

Our results from the *constrained* model ensemble demonstrate that models that show a very strong increase in TXx at the end of the 21st century are unlikely to be realistic. These findings are qualitatively consistent with results by Sippel et al. (2017), who have identified a positive bias in present-day TXx that appears related to a land-surface coupling metric derived from evapotranspiration and temperature; however the metric when applied as constraint could not substantially reduce the spread of projections. Further, Donat et al. (2017) found that an increase in sensible heat and decrease in latent heat flux at the specific day when the hot extreme occurs contributes to strong projected increases of TXx beyond local mean temperatures. When comparing the local scaling of TXx with annual mean temperature of CMIP5 models with observations they find that this in line with observations in CEU. Although this would suggest that the simulated projected changes of TXx are realistic, the models that overestimate TXx might also overestimate annual mean temperature increases, resulting in the same scaling.

Our analysis suggests that most of the *very dry* and *dry* models are unrealistic. This challenges the conclusions of Orth et al. (2016), who suggested that CMIP5 models might show too little drying. However, that study was based on the analysis of a single event, whereas we consider long-term changes of 20-year means.

By comparing these results with projections of Regional Climate Models (RCMs) for Europe (as part of the Coordinated Regional Climate Downscaling Experiment, CORDEX) we find highly inconsistent conclusions. When using observation-based sensible heat fluxes to constrain projections of regional climate models for Europe, (Stegehuis et al., 2013) concluded that summer temperature projections may be underestimated by up to 1 °C regionally in Central Europe . Another RCM-based study suggests that models tend to be prone to a summer temperature bias in Central Europe which cannot be removed with linear bias correction due to non-linear behavior of soil moisture (Bellprat et al., 2013). More recently it has been suggested that in large parts of CEU many of the RCMs tend to overestimate the coupling strength in comparison to observational evapotranspiration products (Knist et al., 2017), which would be in agreement with the behaviour of global CMIP5 ESM simulations according to our findings (and also consistent with (Sippel et al., 2017)). However, the relatively small number of observation time series limits the confidence of their conclusion. The discrepancy between CMIP5 ESMs and RCMs might be largely driven by differences in areosol forcing in the simulations. The RCM CORDEX simulations assume invariant aerosol climatologies, which in turn affects cloud cover, and shortwave radiation variability can thus not be reproduced (Bartók et al., 2017). These effects can have various secondary effects on climate variables such as precipitation and temperature.

## 5 Conclusions and Outlook

In this study we identify a divergent behavior of summer precipitation in long-term projections in Central Europe in a high-emissions multi-model ensemble. The resulting trimodal distribution of precipitation at the end of the 21st century allows a classification into *wet*, *dry* and *very dry* models. The three identified model subgroups are largely overlapping for the next few decades. However, they strongly diverge after global mean warming exceeds 1.3 °C over pre-industrial levels. We find that summer precipitation in the three different model groups is strongly related to latent heat flux and top soil moisture and contributes to large uncertainties in TXx. *Wet*, *dry* and *very dry* models show different behavior, which hints to systematic



differences in the representation of land-atmosphere feedbacks in the models. To understand cause and effect of the detected changes, we investigate model experiments with prescribed soil moisture. The simulations reveal an important role of soil moisture-precipitation feedbacks for the projected precipitation decrease in Central Europe, in addition to the direct effect of soil moisture on temperature. This demonstrates the strong role and complexity of soil moisture feedbacks to the near-surface

atmosphere and in the projected increase of extreme temperatures in CEU. We find no systematic influence of circulation effects, suggesting a minor role of dynamics for explaining uncertainties in long-term projected changes in TXx. We conclude that there are three main positive feedbacks cycles which are relevant for the observed uncertainties in TXx projections: the direct soil moisture-temperature feedback through effects of soil moisture on the partitioning of the turbulent fluxes, the soil moisture-precipitation feedback (which can enhance the projected drying), and soil moisture-radiation feedbacks (which can

induce a further amplification of the surface drying).

By using the correlation between TXx and summer precipitation as a process-based constraint we can exclude *very dry* and most of the *dry* models, resulting in a reduction of 2 °C in TXx in the multi-model median compared to the full ensemble, which corresponds to a reduction of TXx of 20%. Furthermore, the constrained ensemble shows only a minor decrease in summer precipitation (-0.17mm/day) over CEU until the end of the 21st century.

Our study allows for the first time to substantially reduce uncertainties in the projected changes of TXx in Central Europe in ESM simulations based on a process-based constraint. This thus contributes to a better understanding of why models show uncertainties in climate change projections in CEU and offers an approach to provide more informative and reliable projections of changes in summer droughts and heatwaves in this region.




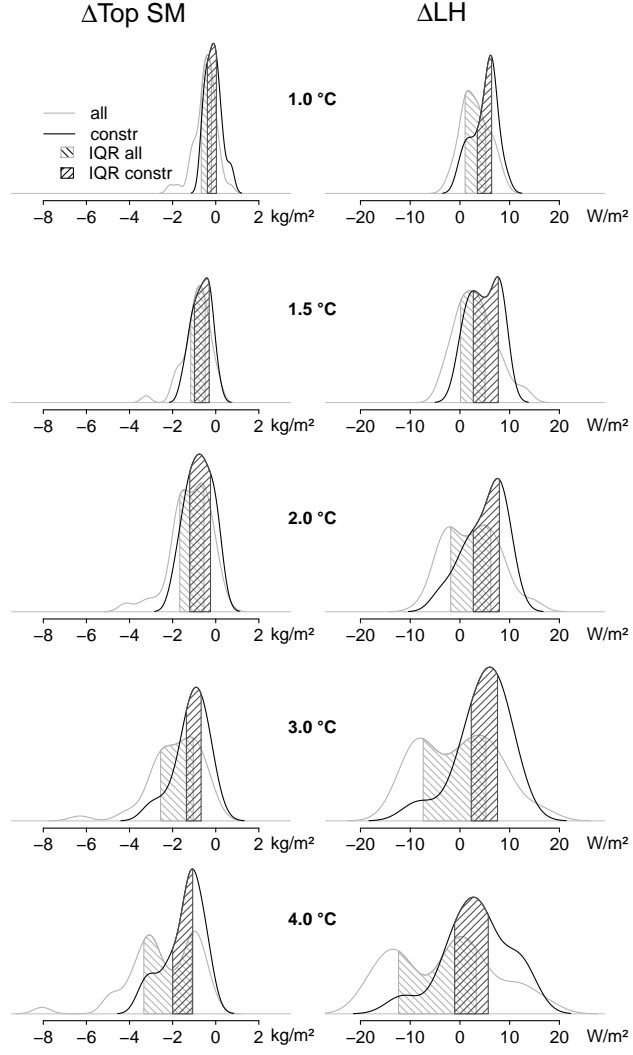

**Figure A1.** Distribution of top soil moisture (top SM, left) and latent heat flux (LH, right) changes according to global warming levels of 1 °C, 1.5 °C, 2 °C, 3 °C and 4 °C for the full (light grey) and the constrained ensemble (dark grey). The shaded area represents the interquartile range (IQR) of the two distributions.

*Data availability.* All used CMIP5 data are available from the public CMIP5 archive. The observational dataset (CRU, EOBS, GPCC, HadEX, Princeton) are available from the respective websites. GSWP3 is available upon request from Hyungjun Kim (hjkim@iis.u-tokyo.ac.jp). The GLACE-CMIP5 data are hosted at ETH Zurich and are available upon request (http://www.iac.ethz.ch/GLACE-CMIP, subject to agreement of the respective modeling groups and database coordinators).



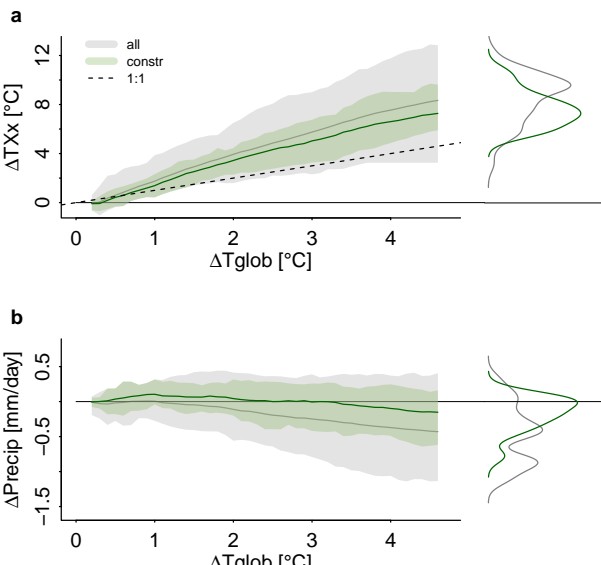

**Figure A2.** Change in TXx and summer precipitation in Central Europe. Changes are 20-year moving means starting from 1951-1970 to 1981-2100. The shaded area shows the minimum and maximum from the whole ensemble (grey) and the constrained ensemble (green).

**Table A1.** Overview of the 23 CMIP5 models. Models marked with * are within the constrained ensemble.

| N° | Model name | Modeling center (or group) | Ensemble member |
|---|---|---|---|
| 1 | ACCESS1.0 | Commonwealth Scientific and Industrial Research Organization (CSIRO) and Bureau of Meteorology (BOM), Australia | r1i1p1 |
| 2 | ACCESS1.3 | Commonwealth Scientific and Industrial Research Organization (CSIRO) and Bureau of Meteorology (BOM), Australia | r1i1p1 |
| 3 | BCC-CSM1.1* | Beijing Climate Center, China Meteorological Administration | r1i1p1 |
| 4 | CanESM2* | Canadian Centre for Climate Modelling and Analysis | r1i1p1, r2i1p1, r3i1p1, r4i1p1, r5i1p1 |
| 5 | CCSM4 | National Center for Atmospheric Research | r1i1p1, r6i1p1 |
| 6 | CESM1(BGC)* | Community Earth System Model Contributors | r1i1p1 |
| 7 | CNRM-CM5 | Centre National de Recherches Météorologiques / Centre Européen de Recherche et Formation Avancée en Calcul Scientifique | r1i1p1 |
| 8 | CSIRO-Mk3.6.0 | Commonwealth Scientific and Industrial Research Organization in collaboration with Queensland Climate Change Centre of Excellence | r1i1p1, r2i1p1, r3i1p1, r4i1p1, r5i1p1, r6i1p1, r7i1p1, r8i1p1, r9i1p1, r10i1p1 |
| 9 | FGOALS-g2* | LASG, Institute of Atmospheric Physics, Chinese Academy of Sciences and CESS,Tsinghua University | r1i1p1 |
| 10 | GFDL-CM3 | NOAA Geophysical Fluid Dynamics Laboratory | r1i1p1 |
| 11 | GFDL-ESM2G | NOAA Geophysical Fluid Dynamics Laboratory | r1i1p1 |
| 12 | GFDL-ESM2M | NOAA Geophysical Fluid Dynamics Laboratory | r1i1p1 |
| 13 | HadGEM2-CC* | Met Office Hadley Centre (additional HadGEM2-ES realizations contributed by Instituto Nacional de Pesquisas Espaciais) | r1i1p1 |
| 14 | HadGEM2-ES | Met Office Hadley Centre (additional HadGEM2-ES realizations contributed by Instituto Nacional de Pesquisas Espaciais) | r1i1p1, r2i1p1, r3i1p1, r4i1p1 |
| 15 | INM-CM4 | Institute for Numerical Mathematics | r1i1p1 |
| 16 | IPSL-CM5A-LR* | Institut Pierre-Simon Laplace | r1i1p1, r2i1p1, r3i1p1, r4i1p1 |
| 17 | IPSL-CM5A-MR | Institut Pierre-Simon Laplace | r1i1p1 |
| 18 | IPSL-CM5B-LR* | Institut Pierre-Simon Laplace | r1i1p1 |
| 19 | MIROC-ESM* | Japan Agency for Marine-Earth Science and Technology, Atmosphere and Ocean Research Institute (The University of Tokyo), and National Institute for Environmental Studies | r1i1p1 |
| 20 | MIROC-ESM-CHEM* | Japan Agency for Marine-Earth Science and Technology, Atmosphere and Ocean Research Institute (The University of Tokyo), and National Institute for Environmental Studies | r1i1p1 |
| 21 | MIROC5* | Atmosphere and Ocean Research Institute (The University of Tokyo), National Institute for Environmental Studies, and Japan Agency for Marine-Earth Science and Technology | r1i1p1,r3i1p1 |
| 22 | MRI-CGCM3 | Meteorological Research Institute | r1i1p1 |
| 23 | NorESM1-M* | Norwegian Climate Centre | r1i1p1 |



**A1**

*Author contributions.* M.M.V., J.Z. and S.I.S. designed the study. M.M. Vogel conducted the analysis. All authors reviewed the manuscript.

*Competing interests.* The authors declare that they have no competing financial interest.

*Disclaimer.* TEXT

5 *Acknowledgements.* We would like to thank Rene Orth for discussions on the initial idea of the study. We thank Alexis Berg, Frederique Cheruy, Stefan Hagemann, David Lawrence, Ruth Lorenz, Arndt Meier and Bart van den Hurk for providing the GLACE-CMIP5 simulations. The research for this study has received funding from the European Research Council (ERC) under grant agreement number 617518 (DROUGHT-HEAT).



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
