# Peer review of "Varying soil moisture-atmosphere feedbacks explain divergent temperature extremes and precipitation projections in Central Europe"

_Earth System Dynamics, 2018_

## Referee Comment (RC1) · S. Hagemann (Referee) · 6 Jun 2018

**Manuscript:** Varying soil moisture-atmosphere feedbacks explain divergent temperature extremes and precipitation projections in Central Europe

**Major remarks**

The authors wrote an excellent paper that relates the differences in CMIP5 projections of temperature extremes (TXx) and precipitation projections over Central Europe to different soil moisture-atmosphere feedbacks in the respective GCMs. By constraining the full model ensemble with observed present-day correlations between summer precipitation and TXx, they were not only able to substantially reduce uncertainties in the projected changes of TXx, but they could also show that the constrained future changes in TXx in Central Europe are about 20% lower than projected by the full CMIP6 ensemble.

I suggest accepting the paper for publication after minor revisions have been conducted.

**Minor remarks**

In the following suggestions for editorial corrections are marked in *Italic*.

p. 2 - line 12
…; *Gudmundsson et al. 2017).*

p. 4 - line 10-11
It is written:
"The choice to use all available realizations was made because we found that the intra-model variability is similar to the inter-model variability for the investigated variables."

However, Table 1 indicates something different, as almost all ensemble members of the same model are located within the same group. Here, IPSL-CM5A-LR is the only exception, having one member among the wet models and all other members in the dry group. Please comment on this!

p. 5 – Sect. 3.1
You describe precipitation changes in mm/day. I suggest including also the respective changes in percentage in brackets.

p. 9 – Fig. 3 caption
It is written:
"Changes for GLACE CTL (violet) and GLACE SM20c (pink) are shown."

It seems that you mixed up violet and pink in the text.

It is also written:
"Density distributions are shown for changes during 2081-2100 (right) for the GLACE CTL (violet) and GLACE SM20c (pink)."

No density distribution are provided!

p. 10 – line 23-24
It is written:
"Until 1990 GPCC, GSWP3 and Princeton show very similar changes in precipitation indicating that the forcing datasets were using the same precipitation."

This statement is not correct. The forcing datasets are bias corrected on a monthly basis with gridded observational datasets: Princeton uses CRU precipitation (http://hydrology.princeton.edu/data.pgf.php), GSWP3 uses GPCC precipitation data.

Both CRU and GPCC are based on station data, where the set of stations may partially overlap.

p. 10 – line 34
…-028), *which* corresponds …

p. 11 – line 9
… projected to *decrease* by …

p. 11 – line 12
… the *projections are slightly* reduced …

p. 12 – Fig. 5 caption
… (orange), *very* dry …

p. 12 – line 13
*We identified* wet …

p. 13 – Fig. 6 caption
Green lines are mentioned, which I cannot identify in the figure. Please correct!

p. 14 – line 14
… lead *to increases* in …

p. 16 – line 14-15
It is written:
… "and enhance incoming shortwave radiation. This can directly decrease latent heat flux …"

Why an enhanced incoming shortwave radiation can directly decrease latent heat flux? Probably you mean *directly increase latent heat flux* as more energy is available at the surface?!

p. 17 – Fig. 9 caption
… (yellow) *feedback loops.*

p. 17 – line 6
… and changes *in* thermodynamics …

p. 18 – line 11
… shows *an* important …

p. 19 – line 16
… Europe, *Stegehuis* et *al. (2013)* concluded …

p. 19 – lines 14-26
For the discussion on RCMs, you may take into account results of Hagemann et al. (2009) who investigated projected changes in GCM and RCM simulations, where both models share almost the same physical packages. They found a stronger warming projected by the GCM in the summer over the Danube and Rhine catchments (representing CEU climate). They explained this difference by the finer resolution of the RCM that leads to a better representation of local scale processes at the surface that feed back to the atmosphere, i.e. an improved representation of soil moisture feedbacks to the atmosphere over the Danube and Rhine catchments.

Hagemann, S., H. Göttel, D. Jacob, P. Lorenz and E. Roeckner (2009) Improved regional scale processes reflected in projected hydrological changes over large European catchments. Climate Dynamics 32 (6), doi: 10.1007/s00382-008-0403-9: 767-781

p. 20 – line 16
*Thus, this* contributes …

---

## Referee Comment (RC2) · S. Ghosh (Referee) · 17 Jul 2018

The authors have performed an excellent analysis on understanding the uncertainty in precipitation projections with soil moisture-atmosphere feedbacks. I thoroughly enjoyed reading the manuscript and I recommend acceptance of the manuscript with very minor revision.

1. The authors have used kernel density estimation to get the trimodal pdf of changes in the projected summer precipitation. I have a small query, what is the impact of selection

of bin size on the shape of the derived distribution. This is clearly a tri-modal case, no doubt, but in my humble opinion, if a K-S test can be performed to just show that the distribution across models differ statistically significantly from uniform distribution and unimodal distributions such as normal and gamma, it may strengthen the claims made by the authors.

2. A minor check, in Table 1, the del LH for wet model, does it have negative sign? I guess it is positive, as I can see from the plots. Kindly recheck.

3. Another minor comment, just to strengthen the conclusions, made by the reviewer, is it possible to statistically show that the classes of very dry, dry and wet models are independent (with the help of multi-variate statistics) when we consider multiple variables, presented in Table 1. This is just a suggestion.

4. Constraining the model with correlations from observation gets rid of extreme models and hence the multi-model projections of summer precipitation shows almost no change. The other way round, probably the models which are not performing well showing extreme and abrupt changes. May be some discussion on this would be a good addition. A small point with this, are we assuming that the correlation will remain unchanged in future? I may be missing something here, but if we are making such assumption, do we have a justification for the same.

5. Fig 9 is an excellent figure summarizing the theory. Just wondering, due to evaporative land surface cooling, is there a possibility of reduction in advective moisture from a distant source?

Finally, this is a fantastic analysis and I am sure this will be a great addition to the literature on understanding the projected climate from models.
* * *

---

## Author Comment (AC1) · 20 Jul 2018

**Response to reviewer 1**

*Responses to reviewer comments are highlighted in italic.*

**Manuscript:** Varying soil moisture-atmosphere feedbacks explain divergent temperature extremes and precipitation projections in Central Europe

**Major remarks**
The authors wrote an excellent paper that relates the differences in CMIP5 projections of temperature extremes (TXx) and precipitation projections over Central Europe to different soil moisture-atmosphere feedbacks in the respective GCMs. By constraining the full model ensemble with observed present-day correlations between summer precipitation and TXx, they were not only able to substantially reduce uncertainties in the projected changes of TXx, but they could also show that the constrained future changes in TXx in Central Europe are about 20% lower than projected by the full CMIP6 ensemble.
I suggest accepting the paper for publication after minor revisions have been conducted.

*We thank the reviewer for his detailed and valuable feedback, which will help us to improve the manuscript. We agree with the content comments and we will correct the grammar/wording as suggested.*

**Minor remarks**
In the following suggestions for editorial corrections are marked in Italic.
p. 2 - line 12
...; Gudmundsson et al. 2017).
*Thanks.*

p. 4 - line 10-11
It is written:
"The choice to use all available realizations was made because we found that the intra-model variability is similar to the inter-model variability for the investigated variables."
However, Table 1 indicates something different, as almost all ensemble members of the same model are located within the same group. Here, IPSL-CM5A-LR is the only exception, having one member among the wet models and all other members in the dry group. Please comment on this!
*Thank you for this comment. We agree with the reviewer and will remove this sentence. We choose all available realizations to have the largest possible dataset.*

p. 5 – Sect. 3.1
You describe precipitation changes in mm/day. I suggest including also the respective changes in percentage in brackets.
*Thank you for this suggestion, we will add relative changes in % in the revised version.*

p. 9 – Fig. 3 caption
It is written:
"Changes for GLACE CTL (violet) and GLACE SM20c (pink) are shown."
It seems that you mixed up violet and pink in the text.
It is also written:

"Density distributions are shown for changes during 2081-2100 (right) for the GLACE CTL (violet) and GLACE SM20c (pink)."
No density distribution are provided!
*Thanks, we will correct this accordingly.*

p. 10 – line 23-24
It is written:
"Until 1990 GPCC, GSWP3 and Princeton show very similar changes in precipitation indicating that the forcing datasets were using the same precipitation."
This statement is not correct. The forcing datasets are bias corrected on a monthly basis with gridded observational datasets:
 Princeton uses CRU precipitation (http://hydrology.princeton.edu/data.pgf.php), GSWP3 uses GPCC precipitation data. Both CRU and GPCC are based on station data, where the set of stations may partially overlap.
*Thank you for this clarification. We will correct this and add a sentence on this in the Data and Methods section 2.3.*

*We thank the reviewer for the suggestions below and implement them.*

p. 10 – line 34
...-028), which corresponds …

p. 11 – line 9
... projected to decrease by …

p. 11 – line 12
... the projections are slightly reduced …

p. 12 – Fig. 5 caption
... (orange), very dry …

p. 12 – line 13
We identified wet …

p. 13 – Fig. 6 caption
Green lines are mentioned, which I cannot identify in the figure. Please correct!
*We meant black lines here, this will be corrected in the revision.*

p. 14 – line 14
... lead to increases in …
*Thanks.*

p. 16 – line 14-15
It is written:
... "and enhance incoming shortwave radiation. This can directly decrease latent heat flux ..."
Why an enhanced incoming shortwave radiation can directly decrease latent heat flux?
Probably you mean directly increase latent heat flux as more energy is available at the surface?!
*Yes, we will change "decrease" to "increase".*

p. 17 – Fig. 9 caption
... (yellow) feedback loops.
*Thanks.*

p. 17 – line 6
... and changes in thermodynamics …
*Thanks.*

p. 18 – line 11
... shows an important …
*Thanks.*

p. 19 – line 16
... Europe, Stegehuis et al. (2013) concluded …
*Thanks.*

p. 19 – lines 14-26
For the discussion on RCMs, you may take into account results of Hagemann et al. (2009) who
investigated projected changes in GCM and RCM simulations, where both models share almost the
same physical packages. They found a stronger warming projected by the GCM in the summer over the
Danube and Rhine catchments (representing CEU climate). They explained this difference by the finer
resolution of the RCM that leads to a better representation of local scale processes at the surface that
feed back to the atmosphere, i.e. an improved representation of soil moisture feedbacks to the
atmosphere over the Danube and Rhine catchments.
Hagemann, S., H. Göttel, D. Jacob, P. Lorenz and E. Roeckner (2009) Improved regional
scale processes reflected in projected hydrological changes over large European catchments.
Climate Dynamics 32 (6), doi: 10.1007/s00382-008-0403-9: 767-781
*Thanks for the comment, we will add this to the discussion.*

p. 20 – line 16
Thus, this contributes …
*Thanks.*

---

## Author Comment (AC2) · 20 Jul 2018

**Response to reviewer 2**
*Responses to reviewer comments are highlighted in italic.*

**Manuscript:** Varying soil moisture-atmosphere feedbacks explain divergent temperature extremes and precipitation projections in Central Europe

The authors have performed an excellent analysis on understanding the uncertainty in precipitation projections with soil moisture-atmosphere feedbacks. I thoroughly enjoyed reading the manuscript and I recommend acceptance of the manuscript with very minor revision.
*We thank the reviewer for his positive feedback and we are pleased that reading of the manuscript was enjoyable. We will incorporate the suggestions in the revision of the manuscript.*

1. The authors have used kernel density estimation to get the trimodal pdf of changes in the projected summer precipitation. I have a small query, what is the impact of selection of bin size on the shape of the derived distribution. This is clearly a tri-modal case, no doubt, but in my humble opinion, if a K-S test can be performed to just show that the distribution across models differ statistically significantly from uniform distribution and unimodal distributions such as normal and gamma, it may strengthen the claims made by the authors.
*It is unfeasible to test against all possible unimodal distributions. Here we use the classification into three model groups rather qualitatively to explain the different feedback processes. Note that the distribution of summer precipitation has no influence on the applied constraint to identify the most realistic models.*

2. A minor check, in Table 1, the del LH for wet model, does it have negative sign? I guess it is positive, as I can see from the plots. Kindly recheck.
*Thank you for the comment. This is indeed true and we will remove the negative sign.*

3. Another minor comment, just to strengthen the conclusions, made by the reviewer, is it possible to statistically show that the classes of very dry, dry and wet models are independent (with the help of multi-variate statistics) when we consider multiple variables, presented in Table 1. This is just a suggestion.
*We are not aware of an approach that would allow us to test this. The independence of climate models is typically very challenging to assess, as many models share similar code and such an assessment would need to go much beyond long-term trends and consider natural variability and the relationships between multiple climate variables.*

4. Constraining the model with correlations from observation gets rid of extreme models and hence the multi-model projections of summer precipitation shows almost no change. The other way round, probably the models which are not performing well showing extreme and abrupt changes. May be some discussion on this would be a good addition. A small point with this, are we assuming that the correlation will remain unchanged in future? I may be missing something here, but if we are making such assumption, do we have a justification for the same.
*The models that agree less with the observations do not necessarily show extreme and abrupt changes. They merely show slightly stronger trends. However, the idea of constraining models with present-day observations indeed includes the strong assumption that the correlation between summer precipitation and TXx does not change in future. Figure 5a shows present-day versus future correlations of summer precipitation and TXx. It can be seen that there is indeed a strong relation. This provides some confidence that our assumption is met. We try to describe this in the Discussion section 4.3 and will add a further comment.*

5. Fig 9 is an excellent figure summarizing the theory. Just wondering, due to evaporative land surface cooling, is there a possibility of reduction in advective moisture from a distant source?
*In our study we did not focus on non-local processes. It might be possible that wet models have stronger local latent heat release and therefore moisture advection, for example from the ocean, is changed. This cannot be answered with our analysis. However, we find it a useful comment and will add a sentence on this.*

Finally, this is a fantastic analysis and I am sure this will be a great addition to the literature on understanding the projected climate from models
*Thank you very much.*